# Stacking Diverse Architectures to Improve Machine Translation

**Andrea Schioppa**                                                                 *arischioppa@google.com*
*Google Research*

**Nal Kalchbrenner**                                                                 *nalk@google.com*
*Google Research*

**Reviewed on OpenReview:** *https://openreview.net/forum?id=XXXX*

## Abstract

Repeated applications of the same neural block primarily based on self-attention characterize the current state-of-the-art in neural architectures for machine translation. In such architectures the decoder adopts a masked version of the same encoding block. Although simple this strategy doesn't encode the various inductive biases such as locality that arise from alternative architectures and that are central to the modelling of translation. We propose Lasagna, an encoder-decoder model that aims to combine the inductive benefits of different architectures by layering multiple instances of different blocks. Lasagna's encoder first grows the representation from local to mid-sized using convolutional blocks and only then applies a pair of final self-attention blocks. Lasagna's decoder uses only convolutional blocks that attend to the encoder representation. On a large suite of machine translation tasks, we find that Lasagna not only matches or outperforms the Transformer baseline, but it does so more efficiently thanks to widespread use of the efficient convolutional blocks. These findings suggest that the widespread use of uniform architectures may be suboptimal in certain scenarios and exploiting the diversity of inductive architectural biases can lead to substantial gains.

## 1 Introduction

Transformers (Vaswani et al., 2017) have become the standard architecture for most natural language tasks and improvements attributed to architectural modifications to the vanilla Transformer do not often generalize across tasks and implementations (Narang et al., 2021). The key component of Transformers is self-attention which is regarded as a useful inductive bias that aggregates a global context through pair-wise interactions between the elements in the sequence. While recent work on architectures (Tolstikhin et al., 2021; Wu et al., 2019; Liu et al., 2021; Tay et al., 2021) is questioning the necessity of self-attention, in general such architectures need to be combined in some way with self-attention to close the ensuing quality gap with the Transformer (Liu et al., 2021; Tay et al., 2021). Both the vanilla Transformer and these alternatives are made of repeated applications of the same layer block, such that all layers have the same structure and self-attention appears as a sub-component in each layer.

From a computational standpoint, operations simpler than self-attention lead to a better computational efficiency while from a theoretical point of view one wishes to better understand why self-attention is often necessary to achieve SOTA results in these language tasks. We propose however an additional motivation to study the effects of simpler operations on language tasks. Different layers can offer different, but complementary inductive biases that can prove useful. For example, convolutions provide a strong bias for local contexts around each element of the sequence, which improves performance in tasks such as speech recognition (Gulati et al., 2020), text summarization (Aksenov et al., 2020) and character-level machine translation (Kalchbrenner et al., 2016). From a historical perspective, locality is one of the main motivations for n-gram language

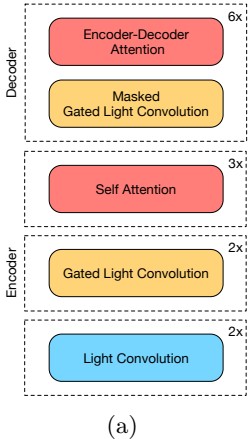
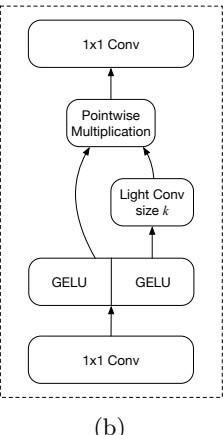

(a)  (b)

Figure 1: (a) Lasagna architecture that stacks diverse architectures to capture multiple inductive biases. (b) Diagram of the Gated Light Convolution layer used in Lasagna.

models and of early neural probabilistic language models (Bengio et al., 2003). Additional evidence comes from modern language models that have access to global context but often default to primarily using the local one (Khandelwal et al., 2018).

We propose to combine complementary inductive biases into a heterogeneous architecture, called Lasagna, to efficiently improve the performance of machine translation models. The architecture is heterogenous in that it does not repeat the same layer block multiple times like prior approaches, but stacks different layers onto each other capturing the respective inductive biases. We analyse the inductive biases of various layers based on two dimensions: the degree of interaction between the elements in the sequence, on the one hand, and the size of the context, on the other. Based on this categorization we propose a new layer block, the Gated Light Convolution. Light Convolutions (LConv), Gated Light Convolutions (GLConv) and Self-Attention (SA) are the three main building blocks of Lasagna and they are stacked on top of each other in sequence. We conjecture that early layers might prefer to focus on a local context whereas later layers might prefer a global one. This suggests stacking layers according to a local-to-global pattern starting with convolutions and ending with self-attention. The sequential stacking of the different blocks in Lasagna allows the model to achieve greater efficiency than homogeneous approaches that channel different operations in parallel within the same block and repeat the block multiple times (Liu et al., 2021; Tay et al., 2021; So et al., 2019). The resulting Lasagna architecture is a carefully designed combination of convolutional and self-attention layers. Despite prior work aiming to merge convolutions and self-attention either in domains other than machine translation (Cordonnier et al., 2020; Yu et al., 2018) or with a different overall purpose (e.g. the local attention of (Yang et al., 2019) or the architecture search in (Fan et al., 2020)), showing that this architectural combination yields a marked improvement in the performance-efficiency trade-off of NMT models is the open problem that Lasagna addresses.

We design and apply Lasagna to machine translation and we see that it outperforms the Transformer on both translation quality and inference speed. We achieve improved results in three different task settings that involve, respectively, autoregressive left-to-right generation, non-autoregressive parallel generation, and the intermediate setting that uses a parallel deep encoder and a left-to-right, but shallow decoder (Kasai et al., 2021). Our results show that Lasagna improves translation quality by up to +0.9 BLEU points depending on language pair and setting, while maintaining an inference speed that is, respectively, up to +30%, up to +40% and up to +60% higher than the baseline Transformer. Ablation studies give additional insight on the contributions of each part of Lasagna.

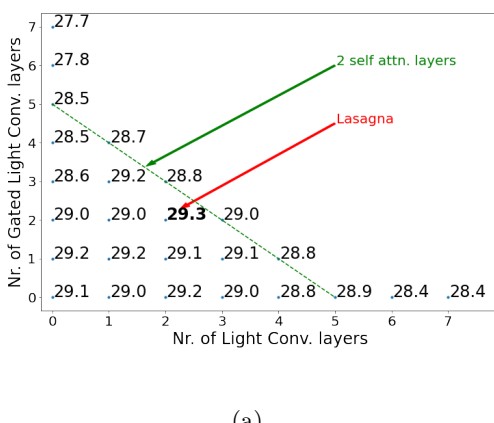
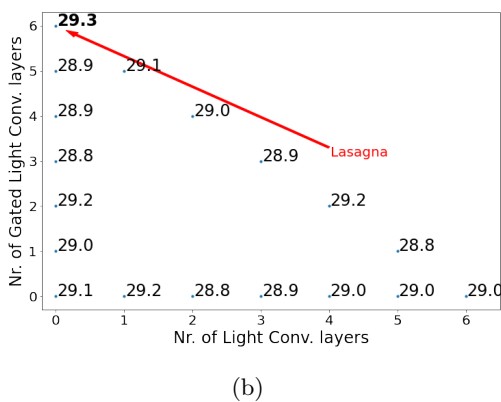

Figure 2: BLEU scores on (EnDe) at model size Big as a function of the number of Light Convolution and Gate Light Convolution layers. Encoder on the left and Decoder on the right. For the Encoder the complement to 7 gives the number of Self-Attention Layers, e.g. 2 for the dashed green line. The variance of scores is 0.1.

## 2 Related work

We summarize related work in different fields. In computer vision, the Mixer architecture (Tolstikhin et al., 2021) proposes an alternative simpler architecture to the SA used in Vision Transformers (Dosovitskiy et al., 2021). Similarly, the more recent gMLP architecture (Liu et al., 2021) outperforms the Mixer on image classification and uses fewer parameters. (Xiao et al., 2021) demonstrates that replacing the standard patchify-stem of Vision Transformers with a standard stride of stacked convolutions improves training stability and can yield gains on classification accuracy. This idea of using convolutions to preprocess the inputs of the Transformer was also considered in Automated Speech Recognition (Mohamed et al., 2019), showing that the positional embeddings are no longer necessary. In machine translation, Kalchbrenner et al. (2016) and (Gehring et al., 2017) introduce a fully convolutional architecture slightly prior to the Transformer; as discussed in Vaswani et al. (2017) however using standard convolutions makes the FLOP count of such models unfavorable compared to that of the Transformer. There have been various attempts at reducing or removing SA starting with Average Attention Networks (AANs) (Zhang et al., 2018), which increase inference speed, while incurring a small drop in translation performance, by replacing the SA in the decoder with an averaging mechanism. This is in line with the findings of (Domhan, 2018) that show that SA is less important in the decoder than in the encoder. Starting from the observation that separable convolutions have a favorable FLOP count to SA, (Wu et al., 2019) introduces two models, Light and dynamic Convolutions, which replace the SA in the encoder and the decoder with layers based on separable convolutions. While both models outperform AANs on inference speed, light convolutions can incur a significant drop in translation quality and in some cases dynamic convolutions are shown to outperform the Transformer. Note that these convolutions are different from standard CNNs, as there is parameter sharing across the heads; moreover, one can also look at SA as a separable convolution where the weight matrix is not a model parameter, but is generated by the query-key pairs. In the following we will refer to convolutions as separable ones. Approaches that channel SA with other operations within the same blocks include (Tay et al., 2021) that proposes a general family of models, called the Synthetizers, that use "synthetic" attention matrices. However (Tay et al., 2021) finds that for machine translation Synthetizers require channel-level combining with SA to close the gap with the Transformer. (Liu et al., 2021) also studies gMLPs on natural language tasks. They consider encoder-only models and do not study generative tasks. While on pre-training perplexity gMLPs match the Transformer, on downstream tasks combining gMLPs with a "tiny" Self-Attention module in the same layer is necessary to close the gap with the Transformer. Enhancing the Transformer by running SA-layers in parallel with convolutional ones has been explored in automated speech recognition (Gulati et al., 2020), machine translation and language modeling (So et al., 2019) and computer vision (Li et al., 2021). Note

that (So et al., 2019) used Neural Architecture Search (NAS) to find the architecture; a large model space is proposed and then pruned by optimizing the model quality on a given task. For a more recent approach to (NAS) that allows each layer to be different see (Fan et al., 2020); however the results of such architecture searches are rarely simple and can be hard to interpret and reuse. Note that we do not run convolutions and SA in parallel in each block, but instead replace SA-layers with simpler and more local operations and stack the operations sequentially. The hybrid strategy of running convolutional layers instead of the MLP layers directly in the SA blocks (Lei et al., 2018) proved not to be competitive in our experiments (see Appendix F). Our work is thus closer to the Sandwich Transformer (Press et al., 2020) that reorders the operations in the Transformer resulting in a non homogeneous stacking strategy. The (Press et al., 2020) uses only SA as a sequence operation and does not improve over the Transformer baseline on machine translation. Finally, hybrid architectures with RNN-decoders and Transformer Encoders were considered in (Chen et al., 2018); moreover (Chen et al., 2018) considered also hybrid Encoders combining RNNs and SA either at channel level or in subsequent layers.

## 3 Model Description

While several operations have been proposed to replace SA, a more thorough categorization of the features of these operations is missing. For example, (Tay et al., 2021) introduces a framework that relies heavily on the notion of self-alignments and thus does not cover operations such as gMLPs (Liu et al., 2021) or light and dynamic Convolutions (Wu et al., 2019). Similarly, (Domhan, 2018) introduces a language to describe models, layers and sequence operations, but treats each sequence operation as a black-box without trying to compare different operations in terms of interaction order, locality or computational complexity. On the other hand, (Liu et al., 2021) suggests that gMLPs have interaction order 2 looking at their polynomial expansion. We start from this observation and look at a sequence operation from two different angles:

1. When building the new representation of a time step, how many interactions appear in the calculation? For example SA requires combining together a query, a key and a value, resulting in an interaction of order 3. LConv directly averages the time steps via a convolution, so the interaction order is 1. We will make this idea more precise in the next subsection.

2. What is the size of the sequential context that can be used to build the new representation for a given time step? For example in LConv this range is related to the size of the kernel, whereas in SA it is naturally global over the whole sequence.

### 3.1 Interaction degree

The interaction degree of a layer is the degree of the polynomial obtained by replacing its non-linearities with the identity function. Let us consider some concrete examples starting with the Multi-Head Attention (Vaswani et al., 2017). The input here is a triplet $(q, k, v)$ of embedded sequences. We use the superscript $h$ to denote the head, and the subscripts $t$ and $c$ to index, respectively, the location in the sequence and the specific channel. The operation is then defined as:

$$\text{Attn}(q, k, v)_{t,c}^h = \sum_\sigma \text{softmax} \left( \sum_\alpha q_{t,\alpha}^h k_{\sigma,\alpha}^h \right) v_{\sigma,c}^h$$

If we replace the softmax with the identity we obtain a linear combination of the monomials $q_{t,\alpha}^h k_{\sigma,\alpha}^h v_{\sigma,c}^h$, leading to the operation having a degree of order 3. Now let us consider LConv (Wu et al., 2019). The input is a single embedded sequence $z$. Let $W_j^h$ denote the convolutional kernel of width $2k+1$ for the head $h$; then:

$$\text{LConv}(z)_{t,c}^h = \sum_{j=-k}^k W_j^h z_{t+j,c}^h,$$

So we see that the operation LConv has order 1.

| Layer | Degree | Local |
|---|---|---|
| Attention | 3 | No |
| Mixer | 1 | No |
| Light Convolution | 1 | Yes |
| Dynamic Convolution | 2 | Yes |
| Random Synthetizer | 1 | No |
| Dense Synthetizer | 2 | No |
| gMLP | 2 | No |

Table 1: Classification of sequence operations by degree and locality.

## 3.2 Locality

Besides the interaction degree, we can analyze operations in terms of the second criterion, that is the size of the context each operation takes as input. A global operation allows all the time steps to interact, while a local one allows only the time steps within a window of size $K$ to interact. Self-attention (SA), Mixer and Synthetizer are examples of global operations, while LConv and Dynamic Convolutions are examples of local ones. Besides capturing a local inductive bias, such operations can benefit from a reduction in the number of memory accesses during the computation, which is one of the bottlenecks of modern accelerators (Ma et al., 2018). We summarize our classification of existing layers in Table 1.

## 3.3 Gated Light Convolutions

The gMLP operation stands out as it achieves an interaction degree of 2 with a cheap point-wise product, and without using additional operations that act along the sequence dimension, as it is the case, for example, in the Dense Synthetizer and the Dynamic Convolutions. The other ingredient of gMLPs is the use of gating through a GELU non-linearity (Hendrycks & Gimpel, 2016). Note that in Table 1 the only degree 2 local operation is the Dynamic Convolution and there is no local analogue for gMLPs. We thus propose to obtain a local version of gMLPs by proposing the Gated Light Convolution (GLConv):

$$\mathrm{GLConv}(z, w)^h_{t,c} = \mathrm{gelu}(z^h_{t,c}) \cdot \mathrm{LConv}(\mathrm{gelu}(w)^h_{t,c}),$$

which is local and has interaction degree 2. Note that while standard gMLPs do not use heads, we partition the channel dimension into heads as in the case of LConv. See Figure 1b for a diagrammatic description of the layer. Note also that in this Section we use the convention of just looking at the sequence operation itself. On the other hand, a full-layer description, requires also to show how $z$ and $w$ are obtained; therefore the full layer description is that in Figure 1b. Specifically, these are obtained applying an inner projection to the input of the layer, similarly in how keys, values and queries are obtained in the case of SA. Similarly, the output of the layer also requires an outer projection as in the case of SA.

## 3.4 Lasagna Architecture

The goal of Lasagna is to stack diverse layers onto each other in order to optimize inductive biases, performance and efficiency. Lasagna's decoder is made up entirely of GLConv layers. The reasons for this are twofold. First, in both autoregressive (AR) and non-autoregressive (NAR) machine translation, the decoder is called repeatedly and therefore its efficiency has an oversized effect on the model's overall efficiency. Lasagna's decoder aims to minimize use of the relatively inefficient SA operation, while using a convolutional operation instead. Previous work (Zhang et al., 2018; Domhan, 2018) also shows that it is possible to simplify the operations in the decoder without hurting overall performance. As a second reason, we found that among the various convolutional variants, GLConv had the highest degree of interaction of 2, while being still relatively cheap to compute with respect to alternatives like Dynamic Convolution.

If we restrict our search by making the assumption that blocks of the same layer follow each other, we can thus conduct an exact search by optimizing the lengths of the blocks of each layer type, see 4.2. The

motivation for using layers with lower interaction order at the beginning is discussed in 4.6. In the rest of this section we will describe the final architecture we found.

Lasagna's encoder is formed by stacking 2 initial LConv blocks, followed by 2 GLConv blocks and then by 3 SA blocks. Besides the efficiency of the convolutional blocks, the encoder also captures a crucial local-to-global inductive bias, whereby the initial layers build up representations for short sequences of tokens in the input sentence and the final SA layers create a fully global representation of the sentence. See Figure 1a for a diagram. In initial experiments we found the local-to-global scheme to perform better than an inverse global-to-local scheme. The Encoder-Decoder attention is still used as in the standard Transformer architecture by letting each decoder layer attend to the last encoder layer (Vaswani et al., 2017). We also experiment with two versions of Lasagna that have more layers in the encoder: Lasagna-9 that has 3 layers of each kind and Lasagna-12 that has 4 layers of each kind.

# 4 Experiments

## 4.1 General setup

To obtain a fair comparison between the performance of various architectures we use the same code base, Fairseq (Ott et al., 2019), for benchmarking all models and, after fine-tuning the training hyper-parameters on the baseline Transformer, we keep the hyperparameters fixed for all architectures (Narang et al., 2021). As recommended by (Marie et al., 2021) we reproduce rather than copy results from published work, we report sacreBLEU[1] (Post, 2018) and we do not use adhoc post-processing steps. For each model we select the best checkpoint based on the perplexity on validation data and use 3 training runs with different seeds to estimate BLEU scores. Except for English to Turkish (EnTr), we find the variance of different runs to be 0.1 BLEU points, while for (EnTr) we estimate it to be 0.3. A complete hyper-parameter setup can be found in the Appendix C. For estimating inference speed we always decode 128 sentences in a single batch 1) to avoid the inference speed artefacts that might arise with very smaller batch sizes such as 1 and 2) to consider a more realistic bulk-inference setting (Kasai et al., 2021). We train models at two scales, Base and Big, as defined in terms of the number of heads and channel dimensions in (Vaswani et al., 2017). Obviously, the parameter count will vary compared to the Transformer when considering a different architecture. We train and evaluate autoregressive Base models on V100 GPUs, whereas the use GPUs A100 for models of size Big. Our code is available on GitHub at `google-research/lasagna_mt`.

## 4.2 Optimizing the stacking pattern

We first fix a 6-layer decoder consisting of GLConv and optimize the stacking scheme in the encoder. We consider the English to German (EnDe) task (see 4.3 for more details) at a model size corresponding to that of Transformer Big, matching the same number of heads and the embedding dimensions. To approximately match the parameter count of Transformer Big, Lasagna uses 7 layers in the Encoder instead of 6, but still stays faster. In the search, we allocate the first $x$ layers to LConv, the following $y$ layers to GLConv and the remaining layers to SA; see grid search space in Figure 2(a). Having two or fewer SA layers decreases the BLEU scores compared to the baseline Transformer. Most models that interleave some LConvs and some GLConvs perform similarly, with BLEU scores in the range $[29.0, 29.3]$. We choose the stacking scheme in Lasagna as it has the highest BLEU score (see Fig. 2(a)). Note also that the plausible assumption that one could just use GLConvs instead of LConv is contradicted by our experimental results as scores degrade more quickly on the line $x = 0$ than the line $y = 0$ as one moves away from the origin.

We then fix the layers in Lasagna's encoder and optimize layer configurations for the decoder. Results are reported in Figure 2(b). As we can match or outperform the Transformer baseline without using SA, we run fewer experiments that include SA and focus instead on the edges of the search space. The horizontal edge corresponds to replacing SA with LConv, the vertical edge to replacing SA with GLConv, while the diagonal edge corresponds to combining different numbers of LConv and GLConv blocks. As expected, in terms of inference speed we observe that it is preferable not to include SA in the decoder search space, see

---

[1]signature in Appendix D

Appendix G for more details. However we do not find a clear pattern in terms of BLEU scores, suggesting that the specific configuration of layers in the Decoder might matter less.

A different approach to optimizing the architecture might have used Neural Architecture Search (NAS), e.g. (So et al., 2019; Fan et al., 2020). However this would have presented some issues:

- **Search costs.** (NAS) can incur high search costs. For example (So et al., 2019) required 200 Google TPUs v2 for the search on the (EnDe) Machine Translation task; for the same task the search in (Fan et al., 2020) required 75 GPUs days and an ad-hoc CPU-GPU collaborative storage.

- **Lack of compute-quality trade-off.** (NAS) assumes that there is a clear optimization objective, for example perplexity in (So et al., 2019) or BLEU in (Fan et al., 2020). However, we are also interested in increasing model speed and using more simple sequence operations. This is more a multi-optimization problem, which requires finding a Pareto frontier of models, each of which is "optimal" after picking a speed/quality trade-off.

- **Architecture complexity.** The outcome of (NAS) might be hard to interpret, especially in the case of diverse layers as in (Fan et al., 2020). We feel that is harder to improve models that are harder to interpret. To make the choice more understandable, we would need to add ad-hoc rules in the search. In our case, using some heuristics, we managed instead to reduce the search to a 2-dimensional space.

### 4.3 Autoregressive Machine Translation

For this task we consider four data benchmarks and two model sizes, where the baselines are, respectively, the Base Transformer and the Big Transformer. As standard benchmarks we consider WMT'14 English to German (EnDe) and English to French (EnFr) as in (Vaswani et al., 2017). We prepare the data using the scripts in the Fairseq translation examples[2]. To test more distant language pairs, that could benefit from more global inductive bias as captured by SA, we consider the WMT'17 Chinese to English (ZhEn) benchmark, following the preprocessing in (Wu et al., 2019; Hassan et al., 2018) (about 20M pairs), and the WMT'17 English to Turkish (EnTr) benchmark following the preprocessing in (Zhang et al., 2018) (about 0.2M pairs). For both benchmarks the evaluation is performed on *newstest17*. Note that as the architecture search was conducted at model size Big on (EnDe), this should not be considered as a test case as we might have overfitted on it. We therefore evaluate on additional language pairs and consider also the model size Base for (EnDe).

On (EnDe) at model size Base (Table 2) we find that our Lasagna architecture improves by 0.4 BLEU points over the Transformer baseline with a 35% higher inference speed. By making the encoder deeper we can improve up to 0.7 BLEU points keeping the same higher inference speed. Note that using LConvs in the encoder and the decoder results in a significant drop of almost 1.5 BLEU points. Lasagna matches Dynamic convolutions on parameter count and inference speed with a gain of +0.2 BLEU points over Dynamic Convolutions. On this task and model size we also considered Mixer (Tolstikhin et al., 2021) and the original gMLP with the "tiny" Self-Attention (Liu et al., 2021). We find that Mixer incurs a significant drop in translation quality, while gMLPs (using the "tiny" SA) can close the gap with the Transformer but incur a big drop in inference speed. We then move to model size Big (Table 4); as said above this should not be considered as a test scenario, but we need still to compare the performance to the other baselines. We observe that models perform comparably on translation quality except for LConvs and Mixer. For Mixer the situation improves at this model size, with just a small drop in translation quality, but there is no significant improvement on the inference speed compared to the baseline. Note that in the case of Lasagna models we can slightly improve the BLEU score (up to +0.4) while increasing the inference speed by about 20%. Note also that at model size Big we did not observe an increased decoding speed by using Dynamic Convolutions. Finally, in both settings we saw that inference speed correlates well with training speed and Lasagna reduces the training time by about 25%. While we evaluate models on BLEU, for (EnDe) at model size Big we also ran an evaluation on model-based metrics, using the framework COMET (Rei et al., 2020); we find that Lasagna and Transformer are on par, while Lasagna(12) outperforms the Transformer, see Appendix A.

---

[2]https://github.com/pytorch/fairseq/examples/translation

| Model | BLEU | Params (M) | tokens/s (k) |
|---|---|---|---|
| Lasagna(12) | **27.2** (+0.7) | 81 | 3.91 (+35%) |
| Lasagna | 26.9 (+0.4) | 67 | 3.92 (+35%) |
| Lasagna(9) | 26.8 (+0.3) | 72 | 3.92 (+35%) |
| Dynamic Conv. | 26.7 (+0.2) | 67 | 3.69 (+28%) |
| Transformer | 26.5 (+0.0) | 66 | 2.88 (+0%) |
| gMLP as in (Liu et al., 2021) | 26.4 (-0.1) | 67 | 1.62 (-44%) |
| Mixer | 25.7 (-0.8) | **65** | 3.71 (+29%) |
| Light Conv. | 25.1 (-1.4) | 66 | **4.12** (+43%) |

Table 2: (EnDe) Base Model. Our Lasagna models outperform the Transformer baseline on Translation quality, while achieving the same inference speed as Dynamic Convolutions. The original gMLP (Liu et al., 2021) architecture with tiny attention is substantially slower than the Transformer. Mixer and Light Convolutions achieve lower BLEU scores than the baseline. For details see 4.3.

For the baseline of the Evolved Transformer (So et al., 2019) we were not able to re-implement it in our codebase. But the authors in Table 3 in (So et al., 2019) report a comparison on the same WMT En-De benchmark that we use after making the training conditions exactly the same for the Transformer and the Evolved Transformer. We can thus compare our gains over the Transformer computing the BLEU score in the same way as in (So et al., 2019). With respect to this Transformer baseline, we achieve a +0.4 gain for Lasagna at model size Base vs a +0.5 gain for the Evolved Transformer; Lasagna also achieves a +0.2 gain at model size Big vs a +0.2 gain for the Evolved Transformer. Note that the baselines for these gains are different as each model is compared to its respective Transformer baseline. Moreover, at model size Big we can report a comparison with NAS search (So et al., 2019; Fan et al., 2020) using BLEU instead of SacreBLEU. At model size Big our Lasagna achieves 30.2 vs 29.8 of the Evolved Transformer, and 30.1 of (Fan et al., 2020). We therefore match the score in (Fan et al., 2020) without an expensive NAS search and a simpler model architecture. We note however that, although presented with the greatest possible care, such comparisons have to be taken with a grain of salt as the baseline is quoted rather than reproduced under the same training conditions (Narang et al., 2021).

For (ZhEn) we observe a similar picture. At model size Base (Table 3) we find that Lasagna and Lasagna-12 achieve +0.4 and +0.8 over the baseline while maintaining a higher decoding speed. At model size Big (Table 5) Lasagna can still outperform the baseline on BLEU score (+0.3) and inference speed (+10%). Here we observe that using LConvs in the Decoder results in a decrease in BLEU.

For (EnFr) and (EnTr) we report the complete experimental results in the Appendix B and summarize here the main findings. For (EnFr), at model size Base we can improve BLEU by +0.9 and inference speed by 30%, while at model size Big we can improve BLEU by +0.3 and inference speed by +28%. For (EnTr), we find that the Transformer achieves a test BLEU of 17.7 while Lasagna-9 achieves 18.2 at comparable inference speed.

We also used bootstrap resampling to estimate the probability that Lasagna scores higher than the Transformer baseline. Overall, we see that these align with the increases in BLEU scores; details can be found in Appendix A.

### 4.4 Non-Autoregressive Machine Translation

For non-autoregressive Machine Translation we took as baseline the Levenshtein Transformer (Gu et al., 2019) using the Fairseq implementation[3]. This model uses the decoder to iteratively refine the translations using a post-editing process learned through imitation learning. We were unable to reproduce the original result of (Gu et al., 2019), where the non-autoregressive model matches the auto-regressive baseline. This was the case both when training with our own distillation data, obtained from our best (EnDe) model at size Big, as well as with the distillation data released with Fairseq; we report results based on the latter. We find

---

[3]https://github.com/pytorch/fairseq/examples/nonautoregressive_translation

| Model | BLEU | Params (M) | tokens/s (k) |
|---|---|---|---|
| Lasagna(12) | **23.3** (+0.8) | 118 | 3.70 (+9%) |
| Lasagna | 22.9 (+0.4) | 104 | 4.01 (+19%) |
| Lasagna(9) | 22.7 (+0.2) | 109 | 3.77 (+12%) |
| Dynamic Conv. | 22.6 (+0.1) | 105 | 3.64 (+8%) |
| Transformer | 22.5 (+0.0) | 104 | 3.38 (+0%) |
| Light Conv. | 22.5 (+0.0) | **100** | 3.98 (+18%) |

Table 3: (ZhEn) Base Model. Our Lasagna models outperform the Transformer and Dynamic Convolutions baselines both on Translation quality and inference speed. For details see 4.3

| Model | BLEU | Params (M) | tokens/s (k) |
|---|---|---|---|
| Lasagna(12) | **29.5** (+0.4) | 277 | 3.77 (+18%) |
| Lasagna | 29.3 (+0.2) | 221 | 3.86 (+21%) |
| Lasagna(9) | 29.1 (+0.0) | 243 | 4.00 (+25%) |
| Transformer | 29.1 (+0.0) | 221 | 3.19 (+0%) |
| Dynamic Conv. | 29.0 (-0.1) | 224 | 3.30 (+3%) |
| Mixer | 28.8 (-0.3) | **206** | 3.14 (-1%) |
| Light Conv. | 28.6 (-0.5) | **206** | 3.22 (+1%) |

Table 4: (EnDe) Big Model. Increasing the scale the differences on Translation quality as measured by BLEU become smaller. Even Mixer and Light Convolutions almost match the Transformer and Dynamic Convolution baselines. Our Lasagna models outperform the baselines in terms of inference speed and BLEU. For details see 4.3.

| Model | BLEU | Params (M) | tokens/s (k) |
|---|---|---|---|
| Lasagna | **24.7** (+0.3) | 296 | 3.78 (+10%) |
| Lasagna(9) | 24.6 (+0.2) | 318 | **3.89** (+13%) |
| Lasagna(12) | 24.6 (+0.2) | 353 | 3.65 (+6%) |
| Transformer | 24.4 (+0.0) | 296 | 3.45 (+0%) |
| Dynamic Conv. | 24.2 (-0.2) | 299 | 3.31 (-4%) |
| Lasagna(9) + LConv Decoder | 24.0 (-0.4) | 312 | 3.79 (+10%) |
| Light Conv. | 24.0 (-0.4) | **281** | 3.46 (+0%) |

Table 5: (ZhEn) Big Model. Our Lasagna models slightly outperform the Transformer baseline on BLEU and both the Transformer and Dynamic Convolutions on inference speed. Note how using Light Convolutions instead of a Light gMLP Decoder results in a -0.4 drop in BLEU. For details see 4.3.

| Model | BLEU | Params (M) | tokens/s (k) |
|---|---|---|---|
| Lasagna(12) | 25.3 (+0.9) | 80 | 5.29 (+39%) |
| Lasagna(9) + LConv Decoder | 24.9 (+0.5) | 71 | 5.56 (+46%) |
| Lasagna(9) | 24.7 (+0.3) | 71 | 5.47 (+44%) |
| Lasagna | 24.6 (+0.2) | 66 | 5.26 (+38%) |
| Dynamic Conv. | 24.5 (+0.1) | 66 | 5.05 (+33%) |
| Transformer | 24.4 (+0.0) | 66 | 3.81 (+0%) |
| Light Conv. | 23.8 (-0.6) | **62** | **5.81** (+52%) |

Table 6: (EnDe) using the Levenshtein Transformer. Our Lasagna models improve Translation quality over the Levenshtein Transformer baseline (Gu et al., 2019) up to +0.9 BLEU points with a +40% increase in decoding speed. For details see 4.4.

| Encoder | Decoder | BLEU | Params (M) | tokens/s (k) |
|---|---|---|---|---|
| Transformer(6) | Transformer(6) | 24.4 (+0.0) | 296 | 3.45 (+0%) |
| Transformer(12) | GLConv(1) | **23.8** (-0.6) | 286 | **10.71** (+210%) |
| Lasagna(12) | GLConv (1) | 23.6 (-0.8) | **274** | **10.69** (+210%) |
| Lasagna(12) | Transformer (1) | 23.5 (-0.9) | 275 | 10.16 (+194%) |
| Transformer(12) | Transformer(1) | 23.5 (-0.9) | 288 | 8.65 (+150%) |

Table 7: (ZhEn), Deep Encoder / Shallow Decoder as in (Kasai et al., 2021). Using a Light gMLP decoder increases decoding speed by 60%. Replacing a Deep Transformer Encoder with a Lasagna one increases decoding speed by 40% without affecting the BLEU score. Note however, how the original Deep Encoder / Shallow Decoder using SA suffers a loss of almost 1 BLEU point to the Transformer baseline. For details see 4.5.

that the Lasagna model outperforms the Transformer also in this setting (Table 6) with improvements up to +0.9 BLEU points and about +40% increase in decoding speed. Note that in the case of the Levenshtein Transformer the decoder is called for a fixed number of iterations on the whole sequence, so the complexity analysis is different from the auto-regressive case; see (Kasai et al., 2021) for details.

## 4.5 Deep-shallow models

Recent work (Kasai et al., 2021) questions whether non-autoregressive Machine Translation is actually faster than auto-regressive one; they propose auto-regressive models with a Deep Encoder and a Shallow Decoder as a faster alternative for larger batch sizes, corresponding to the bulk-inference case. Specifically, they propose a Transformer Encoder with 12 layers and a Transformer Decoder with a single layer. So far we measured speed-ups for models where the encoder and the decoder have comparable depth, and the difference might become negligible when the decoder consists of a single layer. Moreover, a decoder using only local operations might take a considerable hit on Translation quality compared to one using SA. We thus test Lasagna also in this setting.

We observe (Table 7) that replacing the Transformer encoder with Lasagna-12 (with just one layer in the decoder) results in the same BLEU score but an increased inference speed (+60%). Note also that the best BLEU score was not achieved by Lasagna, but by combining a standard Transformer encoder with an GLConv-decoder, i.e. a decoder consisting of a single layer employing a Gated Light Convolution; on the other hand, using a decoder with a single Transformer layer results in a decrease of -0.3.

## 4.6 Ablation studies

We evaluate the impact of various choices that we made in Table 8. We use (EnDe) at model size Base. We first invert the order of the layers in the encoder (Inverted Lasagna) to compare what happens if we decide to

| Encoder | Decoder | BLEU | Params (M) | tokens/s (k) |
|---------|---------|------|------------|--------------|
| Lasagna | GLConv | 26.9 (+0.4) | 67 | 3.92 (+35%) |
| Lasagna | LConv | 26.7 (+0.2) | 66 | 3.87 (+34%) |
| Lasagna, GLConv → LConv | GLConv | 26.7 (+0.2) | 66 | 4.04 (+40%) |
| Dynamic Conv. | Dynamic Conv. | 26.7 (+0.2) | 67 | 3.69 (+28%) |
| Transformer | LConv | 26.5 (+0.0) | **63** | 4.07 (+41%) |
| Transformer | GLConv | 26.5 (+0.0) | 65 | 3.95 (+37%) |
| Transformer | Transformer | 26.5 (+0.0) | 66 | 2.88 (+0%) |
| Inverted Lasagna | GLConv | 26.3 (-0.2) | 67 | 3.97 (+38%) |
| Lasagna, SA → GLConv | GLConv | 26.2 (-0.3) | 66 | 3.71 (+29%) |
| LConv | LConv | 25.1 (-1.4) | 66 | **4.12** (+43%) |

Table 8: Ablations on design choices for the Lasagna. We show that both the order of layers (from local to global) and the use of GLConvs matter. We show that the Transformer Decoder can be replaced by one using GLConvs or LConvs without affecting the BLEU score. For details see 4.6.

first work with global operations and then use local ones. This has a big negative impact as the performance decreases by $-0.6$ BLEU points relative to Lasagna. Secondly we consider what happens if we replace the SA layers in the encoder with GLConvs. In this case the performance also decreases by $-0.7$ points. However, note that in both cases the performance drop compared to the Transformer baseline is small. This ablation suggests that the locality prior of LConv and GLConv complements the global attention prior of SA when LConv and GLConv are used before SA. Both inverting the order of layers in Lasagna as well as removing SA from Lasagna hurt translation quality.

Note that in the (AR) setting the Encoder is called once but the Decoder is called multiple times. In this setting in which the number of layers between the Encoder and the Decoder is the same we found that the inference speed depends on the Decoder only. For example using the Lasagna Encoder or the Transformer Encoder with a decoder using GLConvs results in the same inference speed. Therefore, in this setting the Decoder is responsible for the inference speed gains and the Encoder for the slightly improved translation result ($+0.4$ BLEU points). We thus compared the time for the forward pass in the Lasagna Encoder and the Transformer Encoder for a batch size 64 and found that the former achieves about a $+30\%$ speed increase. The situation is different in the case of the Deep and Shallow models in Table 7 where the Decoder consists of just one layer and the Encoder has 12 layers. In this setting we found the speed-up in the Lasagna Encoder also to have an effect, even when combined with a Transformer Decoder. Finally we warn the reader that the speed measurements might depend on the framework and the computational model. We used Fairseq (Ott et al., 2019) which uses PyTorch thus allowing the batch size to be readjusted during beam search as computations are only done on the hypotheses that need to be continued. The speed is therefore affected also by the number of tokens generated during the search process. On the other hand, frameworks like Tensorflow or JaX use a computational model requiring a fixed batch size, and therefore they do forward passes also on hypotheses that do not need to be continued.

We then look at the role of the GLConvs in the encoder and the decoder. If we replace them with LConvs there is a small impact of $-0.2$ points. Note however that for (ZhEn) at model size Big (Table 5) the impact was larger with a $-0.4$ drop compared to the Transformer baseline (Table 5). Combining a Transformer encoder with a decoder using LConvs or GLConvs does not improve the BLEU score over the baseline.

## 5 Conclusions

We have proposed a novel architecture for machine translation that improves over the Transformer in terms of translation performance and inference speed across a large number of benchmarks and model sizes. The architecture stacks different layers onto each other in order to capture multiple inductive biases and increase performance while increasing inference speed; as a result, most of the self-attention layers that are standard in the Transformer are replaced by lower degree and more local operations. We hope that our work can spur

more interest in this sort of heterogeneous architectures and result in models that are more efficient to train and deploy than the Transformer.

## Broader Impact Statement

The architecture proposed here can reduce the energy consumption for training and deploying Neural Machine Translation Models, as training time is reduced and inference speed is increased. Despite the advances in Machine Translation, such systems may still produce incorrect translations, and this can negatively affect their users. However this negative impact affects Neural Machine Translation in general, and is not specific to the methodology proposed here.

## Acknowledgments

We wish to thank the anonymous reviewers for their insightful comments and suggestions that helped to improve the exposition.

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

## A    Appendix: Significance Testing and Model-based Evaluation.

We used BLEU as the main evaluation metric. In order to assess the significance of the BLEU score differences, we used bootstrap re-sampling to compute the Win Rate between Lasagna and the Transformer Baseline. Specifically, we repeatedly sample subsets of the test set and compute the probability that the BLEU score of Lasagna is higher than that of the Transformer. We feel that using the Win Rate makes the comparison

| Model | Language Pair | Model Size | Win Rate BLEU |
|---|---|---|---|
| Lasagna | EnDe | Base | 85% |
| Lasagna(9) | EnDe | Base | 69% |
| Lasagna(12) | EnDe | Base | 95% |
| Lasagna | EnDe | Big | 87% |
| Lasagna(9) | EnDe | Big | 41% |
| Lasagna(12) | EnDe | Big | 70% |
| Lasagna | ZhEn | Base | 94% |
| Lasagna(9) | ZhEn | Base | 68% |
| Lasagna(12) | ZhEn | Base | 100% |
| Lasagna | ZhEn | Big | 89% |
| Lasagna(9) | ZhEn | Big | 93% |
| Lasagna(12) | ZhEn | Big | 78% |
| Lasagna | EnFr | Base | 77% |
| Lasagna(9) | EnFr | Base | 88% |
| Lasagna(12) | EnFr | Base | 100% |
| Lasagna | EnFr | Big | 72% |
| Lasagna(9) | EnFr | Big | 97% |
| Lasagna(12) | EnFr | Big | 97% |
| Lasagna | EnTr | Base | 86% |
| Lasagna(9) | EnTr | Base | 72% |
| Lasagna(12) | EnTr | Base | 60% |

Table 9: Win Rates for Lasagna over the Transformer for BLEU. Overall the Lasagna systems score higher, i.e. the Win Rate is $> 50\%$.

| Model | wmt20-comet-da | pval | wmt21-comet-mqm | pval |
|---|---|---|---|---|
| Lasagna(12) | **0.3731** | 0.002 | **0.0310** | 0.0005 |
| Lasagna(9) | 0.3622 | 0.2564 | 0.0308 | 0.2329 |
| Lasagna | 0.3585 | 0.56 | 0.0307 | 0.9 |
| Transformer | 0.3553 | - | 0.0307 | - |

Table 10: (EnDe) Big Model. Comparison of Lasagna models with the Transformer on model-based metrics. We report the p-values of a paired t-test with the Transformer scores.

between a system and the baseline more intuitive, as it gives an unbiased estimate of the probability that a system scores higher than the baseline.

The results of this significance estimation are in Table 9. Overall the Lasagna systems score higher than the Transformer counterpart.

For (EnDe) models at size Big we also evaluated on model-based metrics, using the the COMET implementation (Rei et al., 2020). Scores are produced by a model that learns to predict human evaluations. Results are in Table 10 for the metrics `wmt20-comet-da` and `wmt21-comet-mqm`.

## B  Appendix: Tables for English to French and Turkish.

We report our experimental results on (EnFr) at model size Base in Table 11 and Big in Table 13. We report our experimental results on (EnTr) at model size Base in Table 12. Note that Light Convolutions perform

| Model | BLEU | Params (M) | tokens/s (k) |
|---|---|---|---|
| Lasagna(12) | **38.3** (+0.9) | 81 | 3.83 (+30%) |
| Lasagna(9) | 37.9 (+0.5) | 72 | 3.92 (+33%) |
| Dynamic Conv. | 37.9 (+0.5) | 69 | 4.00 (+36%) |
| Lasagna | 37.8 (+0.4) | 67 | 3.76 (+28%) |
| Transformer | 37.4 (+0.0) | 67 | 2.94 (+0%) |
| Light Conv. | 37.2 (-0.2) | **63** | **4.30** (+46%) |

Table 11: (EnFr) Base Model. Our Lasagna models outperform the Transformer on BLEU and inference speed. Light convolutions are also a good baseline, probably because of the closeness between language pairs.

| Model | BLEU | Params (M) | tokens/s (k) |
|---|---|---|---|
| Lasagna(9) | **18.2** (+0.5) | 66 | 3.13 (+4%) |
| Lasagna | 18.0 (+0.3) | **60** | 3.18 (+5%) |
| Lasagna(12) | 17.8 (+0.1) | 75 | 3.25 (+8%) |
| Transformer | 17.7 (+0.0) | **60** | 3.02 (+0%) |
| Dynamic Conv. | 17.5 (-0.2) | **60** | 3.19 (+6%) |
| Light Conv. | 16.8 (-0.9) | **60** | 3.31 (+10%) |

Table 12: (EnTr) Base Model. Differences $\leq 0.3$ on BLEU are not significant. Lasagna(9) improves over the Transformer while Light Convolutions drop by almost 1 BLEU point.

competitively on (EnFr) but experience a significant drop on (EnTr), perhaps because these are more distant language pairs and Turkish word order is less rigid than English.

## C   Appendix: Training Hyper-parameters

We preprocess the data with Fairseq (Ott et al., 2019) using BPE vocabularies with splits and training hyper-parameters reported in Table 14. Note that we sometimes simulate training on a larger number of GPUs by using gradient accumulation steps, which in Fairseq are specified with the `-update-freq` parameter.

There are three dropouts: one for the MLPs, one for attention and one for the convolutional weights. For (EnDe) we use, respectively, 0.3, 0.1 and 0.1; for (EnFr) 0.1, 0.1, 0.1; for (ZhEn) 0.2, 0.2, 0.2; for (EnTr )0.3, 0.1, 0.1.

For the learning rate warmup we use a linear schedule to the peak rate with 4k steps at model size Base and (EnFr) at model size Big, 10k at model size Big for (EnDe) and (ZhEn) and NAT.

| Model | BLEU | Params (M) | tokens/s (k) |
|---|---|---|---|
| Lasagna(9) | **42.1** (+0.3) | 245 | 4.22 (+28%) |
| Lasagna(12) | **42.1** (+0.3) | 279 | 4.13 (+25%) |
| Lasagna | 41.8 (+0.0) | 222 | 4.15 (+26%) |
| Dynamic Conv. | 41.8 (+0.0) | 225 | 3.93 (+19%) |
| Transformer | 41.8 (+0.0) | 222 | 3.30 (+0%) |
| Light Conv. | 41.6 (-0.2) | **207** | 4.20 (+27%) |

Table 13: (EnFr) Big Model. Our Lasagna models improve over the Transformer for BLEU and inference speed.

| Model size | Task | GPUs | Steps | Tok. in Batch | Grad Acc. | Lr schedule | Peak lr | Vocab splits |
|---|---|---|---|---|---|---|---|---|
| Base | (EnDe) | 8 V100 | 120k | 4k | 1 | inverse square root | $5 \times 10^{-4}$ | 32k (shared) |
| Base | (EnFr) | 8 V100 | 120k | 4k | 1 | inverse square root | $5 \times 10^{-4}$ | 40k (shared) |
| Base | (ZhEn) | 8 V100 | 120k | 4k | 1 | inverse square root | $5 \times 10^{-4}$ | 32k (separate) |
| Base | (EnTr) | 4 V100 | 10k | 4k | 4 | inverse square root | $10^{-3}$ | 32k (shared) |
| Big | (EnDe) | 16 A100 | 30k | 3.6k | 4 | cosine decay | $10^{-3}$ | 32k (shared) |
| Big | (ZhEn) | 16 A100 | 40k | 3.6k | 4 | cosine decay | $10^{-3}$ | 32k (separate) |
| Big | (EnFr) | 16 A100 | 120k | 5k | 2 | inverse square root | $5 \times 10^{-4}$ | 40k (shared) |
| NAT | (EnDe) | 8 A100 | 300k | 8k | 1 | inverse square root | $5 \times 10^{-4}$ | 32k (shared) |

Table 14: Hyperparameters for Training. Tok. in Batch is the maximum number of tokens in a batch on a single accelerator, while grad accumulation is the number of gradient accumulation steps to reach a desired number of tokens to update on. When the vocabulary splits are shared it means there is a single shared vocabulary between source and target languages.

The embedding dimensions and the number of heads are the standard ones for model Base and Big. For Base we use 512 embedding dimensions and 8 heads, for Big 1024 embedding dimensions and 16 heads. The kernel sizes for Light and Dynamic Convolutions are like those in (Wu et al., 2019).

## D   Appendix: Inference Hyper-parameters

We use beam size 4. For (EnDe), (ZhEn) and (EnFr) we fine-tuned the length penalty for the Transformer on the validation set and use the same one across all models. We use 0.6 for (EnDe) and (EnFr) and 1.4 for (ZhEn). For (EnTr) we fine-tune the length penalty on the validation set *for each model*, we consider the range 0.4 to 1.8, in increments of 0.1. For sacreBLEU we use the signature `case.mixed+nrefs.1+smooth.exp+tok.13a+v.1.5.1`.

## E   Appendix: Speed Gains and Batch Size

In Figure 3 we plot the relative speed gain over the Transformer measured at different batch sizes. We take the tokens/s of each model relative to the tokens/s of the Transformer. We see that the relative gains can change with the batch size. However, the oscillations tend to be less than 10%. We report 128 to simulate a bulk-inference case in which predictions are batched together.

## F   Appendix: Gating-augmented Feed Forward Layers

We compare the approach using the Lasagna model with an alternative route that enhances SA, motivated by the hypothesis that one should *complement* rather than replace SA; see (Tay et al., 2021; Lei et al., 2018). In (Lei et al., 2018) the feed-forward layers were augmented with recurrent operations, but it was found that increasing translation quality would result into slightly decreasing inference speed. We observe that the recurrent operation of SRUs relies on two main ingredients: 1) usage of two gating operations, 2) a light recurrent operation based on a sort of averaging mechanism (compare with (Zhang et al., 2018)). However, an efficient implemenation of SRUs requires a somewhat-involved ad-hoc CUDA kernel and an adhoc initialization scheme. We thus investigate enhancing the feed-forward layers with the simpler GLConvs, which also uses gating and an averaging mechanism through the convolutions, and has also a straightforward implementation. In order not to slow down too much the decoder, we opt for the following architecture: a Transformer encoder with GLConvs replacing the forward layers and the Lasagna decoder.

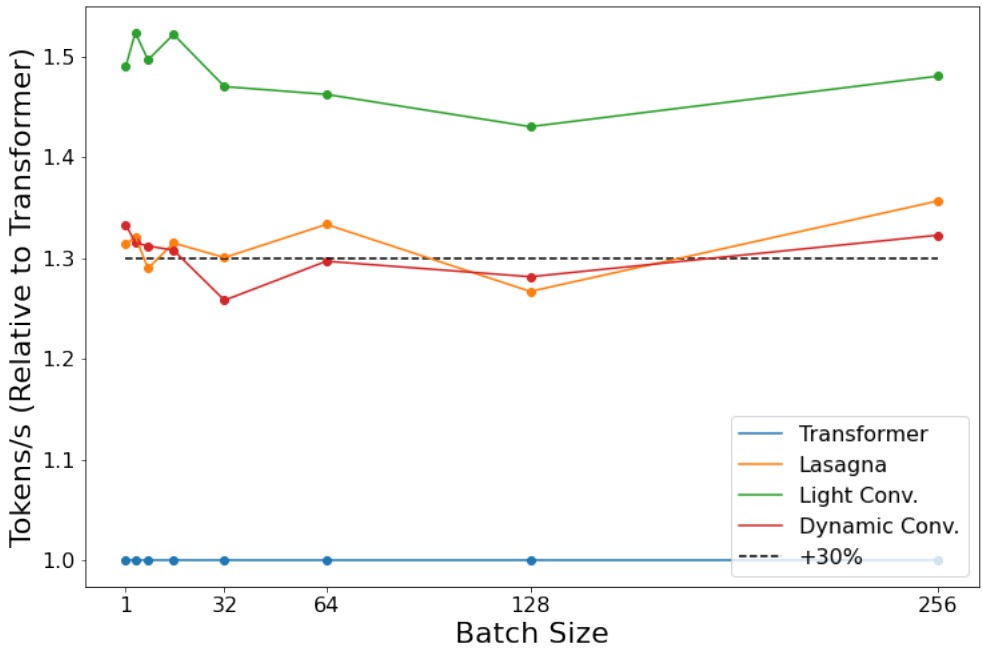

Figure 3: The speed gains over the Transformer do not depend much on the batch size.

| Model | BLEU | Task | Params (M) | tokens/s (k) |
|---|---|---|---|---|
| Lasagna | **24.7** | (ZhEn) | **296** (-12%) | **3.78** (+42%) |
| gating-augmented | 24.6 | (ZhEn) | 337 | 2.60 |
| Lasagna | **29.3** | (EnDe) | **221** (-17%) | **3.86** (+69%) |
| gating-augmented | 29.2 | (EnDe) | 259 | 2.29 |
| gating-augmented | **42.0** | (EnFr) | 260 | 3.39 |
| Lasagna | 41.8 | (EnFr) | **222** (-17%) | **4.15** (+22%) |

Table 15: Our Lasagna model is more effective than augmenting the feed-forwards with Gated Light Convolutions. Less parameters are required to match the same BLEU score and inference speed improves significantly. For details see Appendix F

.

## G    Appendix: Optimizing layers

We report in Figure 4 the results on speed for the layering search for the decoder. The Lasagna encoder already results in a speed gain over the Transformer, even when the decoder uses only SA. However, after removing a few layers of SA most speed results are comparable, given a variance of $\pm 0.1k$ tokens/s.

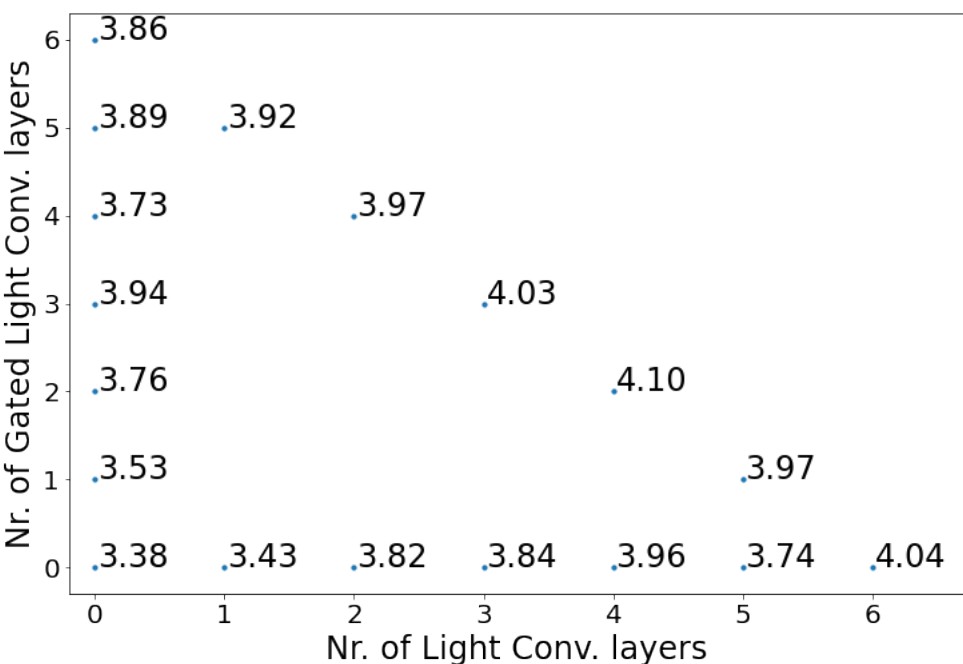

Figure 4: The speed results corresponding to Figure 2(b)

