# OpenReview forum: "Stacking Diverse Architectures to Improve Machine Translation"
_TMLR — Accepted by TMLR_

### Review · Reviewer_3qo6 · 2022-11-12

**Summary Of Contributions:**

For neural machine translation, this paper replaces the transformer encoder by a stack of convolutional and self-attention layers. The convolutions aggregate local information, while the self-attention layers combine global information. In particular, the paper introduces gated light convolutions as a cheap, local and expressive operation. The decoder is also modified to use such convolutional layers.

Over many machine translation datasets and tasks, the proposed architecture can achieve slightly better results than the Transformer while using approximately as many parameters and improving the latency.

**Audience:**

Yes

**Broader Impact Concerns:**

No concerns

**Claims And Evidence:**

Yes

**Requested Changes:**

Clarify that the En-De task should not be considered as a test scenario because architecture search was conducted on it. Critical.

Add additional related work mentioned above. Critical.

More clearly explain the benefits of the proposed changes in the encoder and decoder separately. In Table 8, Transformer encoder + GLConv decoder is faster than Lasagna encoder + GLConv decoder. However, appendix G mentions "The Lasagna encoder
already results in a speed gain over the Transformer, even when the decoder uses only SA.". Critical.

Fix a few typos (p1.suit, p6 outpeforms, p9 BLUE). Not critical.


**Strengths And Weaknesses:**

**Strengths**

The paper covers multiple language pairs (both to/from English), tasks and model sizes.

The proposed approach may improve translation quality while maintaining model size and accelerating inference speed. It is also quite simple.

**Weaknesses**

The architecture search (section 4.2 and figure 2) is conducted on the En-De test set. There is a risk of overfitting to the test data.

The description of gated light convolutions is unclear. How does the equation in section 3.3 relate to figure 1(b)?

Some additional work is missing. In particular, similar work in computer vision and ASR should be cited (e.g. [1] and [2]). Convolutional NMT [3] should also be cited.

[1] Xiao et al. Early Convolutions Help Transformers See Better. 2021. NeuIPS
[2] Mohamed et al. Transformers with convolutional context for ASR. 2019. https://arxiv.org/pdf/1904.11660.pdf
[3] Gehring et al. Convolutional Sequence to Sequence Learning. 2017. ICML

---

> ### Author Response · Authors · 2022-12-16
> **Response to Reviewer 3qo6**
>
> Re: Weaknesses (unclear description of gated light convolutions)
>
> Thank you for the insightful comments. We corrected the equation and explained, below it, how the equation relates to figure 1(b). We hope that this clarifies that, similarly to SA, gated light convolutions use inner and outer projections, as shown in  figure 1(b).
>
> Re: Requested Changes (Clarification of En-De task)
>
> We added the caveat that Big + (EnDe) should not be considered a test case in Section 4.3. We still consider Base + (EnDe) a test case as we did not do hyperparameter search on this configuration. We keep Big + (EnDe) results in Section 4.3 to be able to compare Lasagna to prior baselines.
>
> Re: Requested Changes (Additional related work)
>
> Done. Thanks for pointing it out.
>
> Re: Requested Changes (More clearly explain ...)
>
> Actually we found that timing suspicious and we recomputed the timing for Lasagna + GLConv and the Transformer + GLConv decoder in that setting (updating Tables 2 and 8). We now see that the decoding speed is essentially that of the Decoder. We added a discussion to Section 4.6 (Ablation studies), pointing out that the speed gain in the AR setting is mainly due to the Decoder, while the better Lasagna Encoder can lead to an improvement in translation quality (+0.4 BLEU). We also reported a comparison between the Lasagna Encoder and the Transformer Encoder in the forward pass, showing that the former is faster. This seems to have additional effects both for Non-autoregressive models and for Deep Encoder / Shallow Decoder models.
>
> Re: Typos
>
> Done

---

### Review · Reviewer_XryH · 2022-11-23

**Summary Of Contributions:**

This paper points out that the existing works for solving translation tasks mainly ignore the various inductive biases from different architectures. Therefore, to leverage the advances from different architectures, this paper proposes Lasagna, an encoder-decoder model, which combines convolutional networks and self-attention networks. Experimental results also demonstrate that the proposed Lasagna can outperforms the Transformer baseline.

**Audience:**

Yes

**Broader Impact Concerns:**

There is no concerns here.

**Claims And Evidence:**

Yes

**Requested Changes:**

1. The biggest problem of this problem is its novelty. Many works have investigate the locality problem in machine translation, and introduces different designs by combining transformer and cnn layers. Authors should provide convincing justifications to prove the necessity of the proposed architectures.
2. I want to check one detail: why Lasagna(12) has more parameters than Lasagna, but achieves faster speed? And authors argue that using a deeper encoder can achieve better performance with a higher speed (Page 6). This is really counterfactual and can authors check it again?

**Strengths And Weaknesses:**

**Strengths**
1. This paper is well-written.
2. This paper introduces a well-designed encoder-decoder architecture for solving neural machine translation tasks.

**Weaknessnes**
1. Although some significant progress has been achieved, the novelty of this paper is still limited. Combining CNN and Transformer is not an interesting idea, and has been adopted in many works [1][2][3][4] (including manual design and automatic design). Therefore, it is not surprising that combining self-attention with cnn (lightConv or Gated LightConv) could achieve better results. Locality in machine translation or other generation tasks also has been studied in many works [5], which may also reduce the contribution of this paper.
2. Just as aforementioned, the design of Lasagna is completely hand-crafted. It lacks enough insights or motivation to prove why such a design is optimal. For example, considering the components used in this paper as a search space (LConv, gated LConv and Self-attention), can we use NAS to obtain a better architecture?
3. The improvements are marginal, which is not satisfactory. For example, In EnDe base and Large setting. Lasagna achieves 26.9 (+0.4) and 29.3 (+0.2) over the Transformer baseline, which are not significant.

[1] Searching Better Architectures for Neural Machine Translation

[2] Convolutional self-attention networks

[3] ON THE RELATIONSHIP BETWEEN SELF-ATTENTION AND CONVOLUTIONAL LAYERS

[4] QANet: Combining Local Convolution with Global Self-Attention for Reading Comprehension
[5] Convolution sequence to sequence learning

---

> ### Author Response · Authors · 2022-12-16
> **Response to Reviewer XryH**
>
> Re: Weaknesses 1
>
> Thank you for the insightful review. We include these references and a respective discussion in the paper. In detail, [1] does not use the same building blocks as Lasagna, as Lasagna uses modern separable convolutions, whereas [1] doesn’t; further, the application of NAS in [1] results in a much more complex architecture than Lasagna that is hard to interpret, port or improve. [2] simply restricts the context of their self-attention without explicitly using convolutional operations; for this reason [2] cannot make use of the computational efficiency advantage of convolutions. [3] focuses on on the different domain of computer vision, where the authors aim to approximate a convolutional Resnet with self-attention, and not the other way around as in Lasagna. [4] focuses on the different task of QA and enhances the self-attention layers with convolutions without worrying about reducing the amount of self-attention or increasing the efficiency of the model. Finally [5] doesn’t use separable convolutions, but standard ones, resulting in a much slower model (up to 10-100x the FLOPS of the Transformer) and substantially lower performance than the Transformer. Fully showing the benefit of convolutions in a Transformer architecture while avoiding its advantages is the open problem that Lasagna addresses.
>
> Re: Weaknesses 2
>
> We explain in section 4.2 why we do not perform NAS. NAS can cost a lot to train, usually disregards the inference speed of the models, and results in interpretable and impractical architectures.
>
> Re: Weaknesses 3
>
> To measure significance, we explain in detail in Appendix A an additional estimation of the Win Rate between Lasagna and the Transformer baseline. Bootstrap resampling can distinguish between Lasagna and the Transformer baseline with Win Rates strictly greater than 50%, suggesting a significant improvement.
> In addition, Lasagna’s relative gains are similar to those, for example, of the Evolved Transformer (https://arxiv.org/pdf/1901.11117.pdf, page 8 Table 3) which is +0.5 and +0.2 points better in the Base and Big configurations, respectively.  Both Lasagna and the Evolved Transformer obtain these gains over a highly fine tuned reimplementation of the Transformer baseline. Further, Lasagna achieves these quality gains while being up to 30% faster at inference
>
> Re: Requested Changes 1
>
> Lasagna proposes a simple-to-implement approach to improve over the Transformer baseline on the quality-efficiency trade-off in machine translation. Other works do not offer such a proposal or do not study its results in detail.
>
>
> Re: Requested Changes 2
>
> Thank you for this point. We rechecked the result. In AR the Encoder is called just once, so the inference speed is determined by the Decoder architecture. Both Lasagna and Lasagna(12) use the same Decoder so they should be expected to have the same speed (up to measurement variance). We reran the code and updated the Table ( also in light of Review #3).

---

### Review · Reviewer_8B11 · 2022-12-05

**Summary Of Contributions:**

This paper proposes a non-uniform variant of the transformer architecture, where the lower layers focus on learning local interactions using convolutional layers and the upper layers learn global interactions using self-attention layers. This hybrid architecture allows the model to have efficient training and inference. Experiments on machine translations demonstrate the advantage of the proposed architecture in accuracy and speed.

**Audience:**

Yes

**Broader Impact Concerns:**

None observed.

**Claims And Evidence:**

Yes

**Requested Changes:**

It would be nice to add discussions regarding the two points in "weakness" above, but this is not critical to my recommendation.

**Strengths And Weaknesses:**

Strength:
- The paper is written with clarity.
- The empirical results are strong and detailed analysis are provided to demonstrate the utility of each design.

Weakness:
- It's unclear if the proposed architecture is specific to MT, or it can be used as a general architecture for pretraining (like Transformers).
- It would be nice if the related work engages with the NAS literature.

Overall, this is a solid paper and I'd recommend acceptance.

---

> ### Author Response · Authors · 2022-12-16
> **Response to Reviewer 8B11**
>
> Re: Requested Changes
>
> Thank you for the insightful review. Tasks other than machine translation with the proposed architecture are currently outside of the scope of the paper and we leave it to future work to explore them, although we believe it is a promising avenue indeed. Regarding NAS, we have added a discussion in the related work section of the following NAS papers: "Evolved Transformer” and "Searching Better Architectures for Neural Machine Translation’’.

---

### Decision · Action_Editors · 2023-01-11

**Recommendation:** Accept with minor revision

**Comment:**

This paper explores new architectures for machine translation. The main insights are convolution for local interactions and self-attention for global interactions. The review and discussion processes are good: reviewers have provided good comments and suggestions, and the authors have actively replied and addressed most of these problems. Two reviewers lean to accept and one reviewer leans to reject.

Some reviewers mentioned that it lacks insights and motivation to demonstrate the benefits of this kind of architectural design, improvements over baselines are not large enough, and also comparisons with some previous architecture-design work (e.g., Evolved Transformer) on machine translation are missing. Suggest the authors could add sufficient discussions to clarify the insights and motivation of the architecture design, and compare with more baselines if possible in the revised version.

**Audience:**

Yes, some audiences may have interests.

**Claims And Evidence:**

Most claims are supported.

---

> ### Author Response · Authors · 2023-02-07
> **Added baselines**
>
> Thanks!
> We hope to have addressed the requested changes as follows:
> 1. We added a comparison to the Evolved Transformer and the model from ``Searching better architectures for neural machine translation'' on page 8.
> 2. We elaborated on the motivation for the search strategy in Section 3.